# Nonlinear feedforward enabling quantum computation

**Atsushi Sakaguchi** [1,2] ✉, **Shunya Konno** [1], **Fumiya Hanamura** [1], **Warit Asavanant** [1,2], **Kan Takase** [1,2], **Hisashi Ogawa** [1], **Petr Marek** [3], **Radim Filip** [3], **Jun-ichi Yoshikawa** [1,2], **Elanor Huntington** [4], **Hidehiro Yonezawa** [5] & **Akira Furusawa** [1,2] ✉

Measurement-based quantum computation with optical time-domain multiplexing is a promising method to realize a quantum computer from the viewpoint of scalability. Fault tolerance and universality are also realizable by preparing appropriate resource quantum states and electro-optical feedforward that is altered based on measurement results. While linear feedforward has been realized and become a common experimental technique, nonlinear feedforward was unrealized until now. In this paper, we demonstrate that a fast and flexible nonlinear feedforward realizes the essential measurement required for fault-tolerant and universal quantum computation. Using non-Gaussian ancillary states, we observed 10% reduction of the measurement excess noise relative to classical vacuum ancilla.

A quantum computer promises to solve certain computational tasks significantly faster than a modern computer does. The goal of the research on quantum computing is to realize a practical quantum computer that is scalable, universal and fault tolerant. Various physical systems[1]—e.g., superconducting devices[2,3], trapped-ion systems[4,5], and semiconductor systems[6,7]—have been investigated extensively. Among them, optical systems have unique potential regarding the scalability[8–11]. For example, generation of a large-scale entangled states, the so-called cluster states, has been demonstrated using time-domain multiplexing methods[12–14]. The cluster states are resources of measurement-based quantum computation (MBQC), where quantum operations are performed via local measurements on the large-scale cluster states and feedforward operations depending on the measurement outcomes[15,16]. The demonstrated large-scale cluster states are categorized as continuous-variable cluster states, which treats continuous-valued quadratures $\hat{x}$ and $\hat{p}$ of an electro-magnetic field that satisfy $[\hat{x},\hat{p}] = i$. In continuous-variable MBQC, homodyne measurement is one of the most fundamental and powerful

measurement[17]. When combined with ancillary states and feedforward, homodyne measurement has an ability to implement fault-tolerant universal quantum computation[18–20]. For example, this combination can implement Clifford operations or Gaussian operations (Fig. 1a), error recovery operations (Fig. 1b), and fault-tolerant non-Clifford operations (Fig. 1c). It is, however, emphasized that ancillary states and feedforward must be specifically customized for each operation[18]. In previous research, only deterministic Gaussian operations on Gaussian and non-Gaussian states have been demonstrated with Gaussian ancillary states and solely linear feedforward[21–23]. Deterministic non-Gaussian operations have not been realized so far.

The difficulty of implementing deterministic non-Gaussian operations on traveling optical states stems from the requirement of complicated non-Gaussian ancillae and nonlinear feedforward — conditional Gaussian operations controlled by the nonlinear function of the measurement outcomes. The ancillary states are crucial resources to obtaining quantum non-Gaussianity, and the adaptive control of nonlinear feedforward is essential for deterministic operations. While

[1]Department of Applied Physics, School of Engineering, The University of Tokyo, 7-3-1 Hongo, Bunkyo-ku, Tokyo 113-8656, Japan. [2]Optical Quantum Computing Research Team, RIKEN Center for Quantum Computing, 2-1 Hirosawa, Wako, Saitama 351-0198, Japan. [3]Department of Optics, Palacký University, 17. listopadu 1192/12, 77146 Olomouc, Czech Republic. [4]Centre for Quantum Computation and Communication Technology, School of Engineering, College of Engineering Computing and Cybernetics, Australian National University, Canberra ACT 2600 ACT, Australia. [5]Centre for Quantum Computation and Communication Technology, School of Engineering and Information Technology, University of New South Wales, Canberra ACT 2600 ACT, Australia. ✉e-mail: atsushi.sakaguchi@riken.jp; akiraf@ap.t.u-tokyo.ac.jp

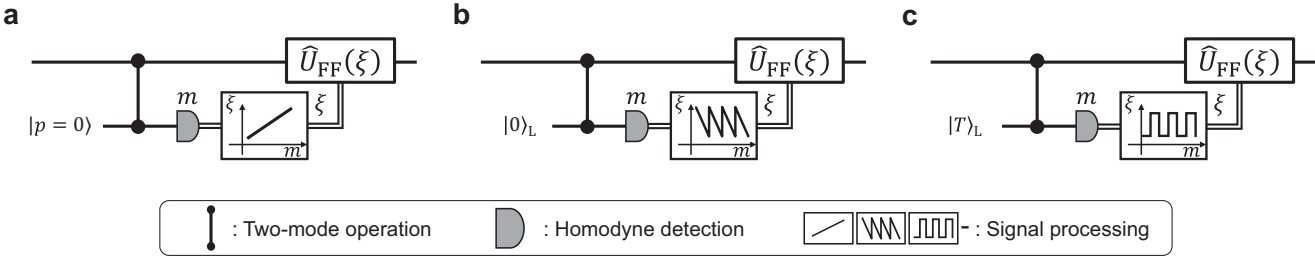

**Fig. 1 | Quantum operations implemented in measurement-based quantum computation. a** Clifford (Gaussian) operation via linear feedforward and a Gaussian ancillary state. **b** Error recovery operation of GKP encoding via nonlinear feedforward and a GKP logical state as an ancilla. **c** Gate teleportation scheme for a fault-tolerant non-Clifford operation via nonlinear feedforward and a magic ancillary state.

the preparation of ancillary states in optical systems has been extensively researched theoretically[24–27] and experimentally[28,29], the development of essential nonlinear feedforward has remained limited. There are a few reports about nonlinear feedforward such as digital feedforward with a primitive digital logic[30] or analog feedforward with dedicated circuits for a specific task[31]. The nonlinear feedforward in the previous researches is, however, inflexible or too slow to synchronize electrical signals and optical signals, which is imperative for MBQC in time domain. Hence, nonlinear feedforward is a key piece to unlock the full potential of an optical quantum computer.

Here, we demonstrate flexible and fast nonlinear electro-optical feedforward and use it to implement a nonlinear quadrature measurement that, in combination with a suitable ancilla, projects the state of traveling light into a non-Gaussian state, as is required for quantum computing. Our setup (Fig. 2a) measures the nonlinear combination of two quadratures $\hat{x}$ and $\hat{p}$ of an electromagnetic field, $\hat{p} + \gamma\hat{x}^2$ where $\gamma$ is a parameter we can tune. This nonlinear quadrature measurement can be readily applied to a non-Clifford operation if combined with the cluster states already demonstrated in refs. 13, 14, 21, 22. Moreover, the nonlinear quadrature measurement is useful for a wide variety of continuous-variable quantum protocols, such as Schrödinger's cat state generation[32], and high-fidelity quantum teleportation[33]. We perform tomography of the tailored measurement, observe 10% reduction of excess noise thanks to the non-Gaussian ancilla, and verify its quantum non-Gaussian nature. The results signify that our feedforward system works properly and the nonlinear quadratures are indeed measured. The nonlinear feedforward system developed in this work is flexible and capable of implementing various signal processing. Therefore, it is applicable not only to specific non-Gaussian operations, but also to fault-tolerant non-Clifford operations on Gottesman-Kitaev-Preskill (GKP) qubits[18], continuous-variable gate teleportation[19] and analog error correction of GKP qubits[19,20], if appropriate ancillary states are prepared. This work has opened a new nonlinear regime beyond large-scale cluster states and Gaussian operations, establishing an important cornerstone of optical quantum computation.

## Results
### Experimental setups
Figure 2b shows a schematic diagram of the experimental system. An input state is interfered with an ancillary state on a beam splitter, and one of the outputs is measured by a homodyne detector (HD1). The measured value $q$ of the homodyne detector is processed by nonlinear functions on a field programmable gate array (FPGA) board. During this signal processing, the other output of the beam splitter is on hold in an optical delay line (ODL2). The calculated results by the FPGA are fed forward to the other homodyne detector (HD2) and set the measurement basis to $\hat{p}_{\theta(q)} = \hat{p}\cos(\theta(q)) + \hat{x}\sin(\theta(q))$ via phase rotation $\hat{R}(\theta(q))$ of the local oscillator beam of HD2. The exact form of the nonlinear feedforward is determined by $\theta(q) = \arctan(\sqrt{2}\gamma q)$ as shown in Fig. 2d. To measure the nonlinear quadrature of $\hat{p}_{in} + \gamma\hat{x}_{in}^2$, the

measurement outcome of the second homodyne detector, $y$, is multiplied by the gain $\sqrt{2}/\cos(\theta(q))$ which is determined by the measurement outcome $q$ of the first homodyne detector. Finally, the outcome $m = \sqrt{2}y/\cos(\theta(q))$ is obtained. This $m$ corresponds to the nonlinear quadrature $\hat{m}$,

$$\hat{m} = \hat{p}_{in} + \gamma\hat{x}_{in}^2 + \left(\hat{p}_{anc} - \gamma\hat{x}_{anc}^2\right) \tag{1}$$

where $\hat{x}_{in}, \hat{p}_{in}$ are quadratures of the input state, and $\hat{x}_{anc}, \hat{p}_{anc}$ are quadratures of the ancillary state. Equation (1) shows that the nonlinear quadrature of the input state, $\hat{\delta}_{in} = \hat{p}_{in} + \gamma\hat{x}_{in}^2$, is influenced by an excess noise caused by the corresponding nonlinear quadrature of the ancillary state, $\hat{\delta}_{anc} = \hat{p}_{anc} - \gamma\hat{x}_{anc}^2$. Note that the excess noise is independent from quadratures of the input state. Hence, the amount of excess noise is determined only by the ancillary state. The ideal ancillary state that gives $\hat{\delta}_{anc} = 0$ is a cubic phase state (CPS), which satisfies

$$\hat{\delta}_{anc}|CPS\rangle = 0. \tag{2}$$

An ideal CPS is an unphysical state because it requires infinite energy to generate. Thus, we must consider an approximated CPS similar to squeezed states substituted for ideal quadrature eigenstates in continuous-variable quantum computation. We call the approximated cubic phase state as a nonlinearly squeezed state or cubic squeezed state[34], since the variance of the nonlinear quadrature operator $\hat{\delta}_{anc}$ is squeezed beyond the lower bound imposed by Gaussian states and their mixtures[35]. It is known that a superposition of photon number states can be a good approximation of nonlinearly squeezed state even with a moderate number of photons in the state[35,36]. Figure 2c shows the Wigner function of the ancillary state used in our experiment. This ancillary state is nonlinearly squeezed by about 10% beyond any Gaussian states or their mixtures when $\gamma = 0.52$ and it has clear regions of negativity. Note that the level of the nonlinear squeezing depends on the coefficient $\gamma$, and this value is optimal for the experimental ancillary state. The ancillary state can be further improved by increasing the number of the photons to generate larger nonlinear squeezing[19,37].

Fast nonlinear feedforward system is a key technological component in our challenging experimental setup. This is because slow signal processing leads to a long optical delay line which entails adverse effects such as loss and phase fluctuation. To implement fast and flexible signal processing, we used an FPGA board equipped with low-latency AD/DA converters and implement a look-up table inside the FPGA. The target function (arctangent) is pre-calculated and stored in the look-up table so that the calculation is accurately completed within 1 clock cycle, 2.67 ns in this experiment. The total latency of the FPGA board is 26.8 ns, corresponds to about 8 meters of optical delay lines which can be feasibly stabilized in experimental setups. This latency does not depend on the form of processing as the values of look-up tables are calculated in advance, thus the feedforward system has significant flexibility.

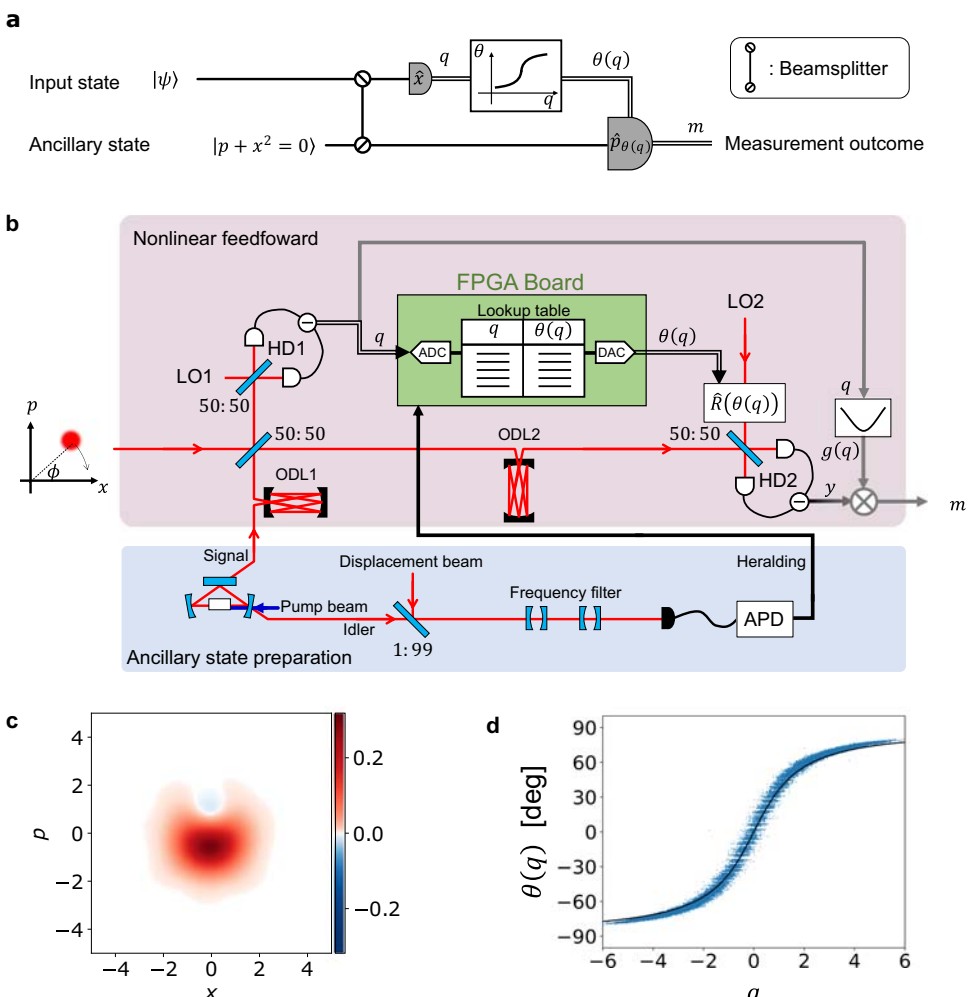

**Fig. 2 | Nonlinear quadrature measurement setup. a** Quantum circuit diagram of the nonlinear quadrature measurement. **b** Schematic of the experimental setup. Input states and ancillary states are localized in time domain. The whole system consists of two parts, a setup for ancillary state preparation via heralding method and a setup for nonlinear feedforward operation. In the ancillary state preparation, an optical parametric oscillator (OPO) is pumped by a frequency doubled beam, generating a two-mode squeezed state. One of the modes (idler mode) is displaced and passed through frequency filters before it is measured by an avalanche photodiode (APD). Click events by the APD herald the success of non-Gaussian ancillary states preparation in the signal mode and trigger the nonlinear feedforward operations. LO local oscillator, HD homodyne detector, ODL optical delay line, FPGA field programmable gate array, ADC analog-to-digital converter, DAC digital-to-analog converter, APD avalanche photodiode. **c** Wigner function of the non-Gaussian ancillary state used in this experiment. **d** Input-output relation of the nonlinear calculation on the FPGA board. The target function, $\theta(q) = \arctan(\sqrt{2}\gamma q)$ where $\gamma = 0.52$, is shown as a black line. Blue dots show experimentally obtained values.

## Tomography of tailored measurement

To experimentally characterize our nonlinear measurement, we input various coherent states $|\alpha\rangle$, where $\alpha$ is the complex amplitude written by two real values $\alpha_x, \alpha_p$ as $\alpha = (\alpha_x + i\alpha_p)/\sqrt{2}$. The input coherent states are carefully calibrated by dual homodyne measurement[17], which is implemented by the same experimental setup with the feedforward system turned off. We choose 27 different amplitudes $|\alpha|$ equally spaced and ranging from 0 to 3.5. The coherent states are sampled with randomized phase in each fixed amplitude $|\alpha|$ (see Supplementary Note 1).

In the Heisenberg picture, the quality of the measurement can be analyzed by looking at the first and second moments of the measured nonlinear quadrature in Eq. (1) for the set of sample coherent states. We indeed saw that the value obtained by the nonlinear measurement matches the theoretical predictions, is unbiased both in the mean and the variance, and that the added noise is determined by the nature of the ancillary quantum state (see Supplementary Notes 2 and 3). In addition, for comprehensive characterization of a quantum measurement, we consider in Schrödinger picture.

The ideal measurement of nonlinear quadrature in Eq. (1) with zero excess noise projects the measured field into the nonphysical displaced cubic phase state in Eq. (2). The practical realization of Fig. 2b projects the field into unnormalized quantum states, which we call detector states in this paper. These unnormalized detector states are also known as positive operator valued measure elements. This projection determines both the probability of obtaining the particular measurement result $m$ and the quantum state prepared in the case when the measurement was applied to one part of a maximally entangled state. In this experiment, the detector state associated to a particular measurement outcome $m$ is

$$\hat{\Pi}_m \propto \int dq \, \hat{U}(q,m)\hat{T}\hat{\rho}_{\text{anc}}\hat{T}^\dagger\hat{U}^\dagger(q,m) \tag{3}$$

$$\hat{U}(q,m) = \exp\left[i\tan\theta(q)\hat{x}^2\right]\hat{D}(\sqrt{2}q, m - \sqrt{2}q\tan\theta(q)) \tag{4}$$

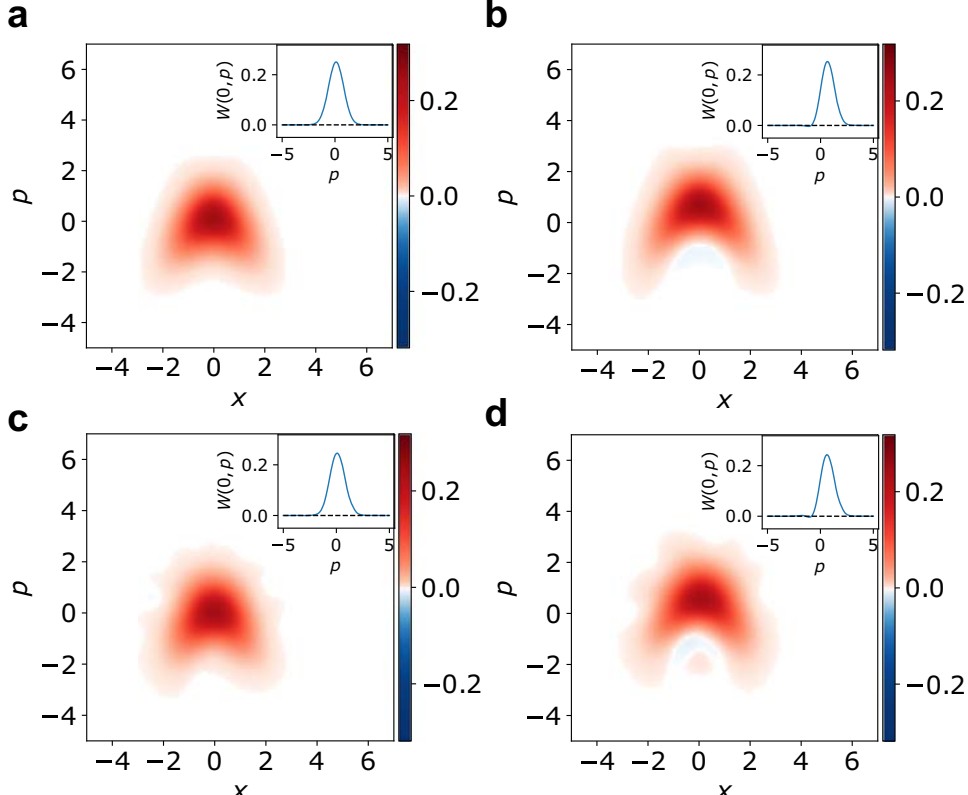

**Fig. 3 | Averaged detector states of the nonlinear quadrature measurement.**
**a** Theoretical detector state for vacuum ancilla with $\mathrm{var}(\hat{p} + \gamma\hat{x}^2) = 0.64$.
**b** Theoretical detector state for non-Gaussian ancilla with $\mathrm{var}(\hat{p} + \gamma\hat{x}^2) = 0.56$.
**c** Experimental detector state for vacuum ancilla with $\mathrm{var}(\hat{p} + \gamma\hat{x}^2) = 0.74 \pm 0.01$.
**d** Experimental detector state for non-Gaussian ancilla with

$\mathrm{var}(\hat{p} + \gamma\hat{x}^2) = 0.67 \pm 0.01$. The detector states are averaged with $p$-displacement by the measurement outcomes $m$ and renormalized. Note that there are ripples in (**c**) and (**d**) that are not present in (**a**) and (**b**), which are artifacts of the reconstruction method (see Supplementary Note 5). The insets show the cross section of Wigner functions along $x = 0$.

where $\hat{\rho}_{\mathrm{anc}}$ is the density matrix of the ancillary state, $\hat{D}(x_0, p_0) = \exp[i(p_0\hat{x} - x_0\hat{p})]$ is a displacement operation, and $\hat{T}$ is an anti-unitary operator represented by $\hat{T}^\dagger\hat{x}\hat{T} = \hat{x}$ and $\hat{T}^\dagger\hat{p}\hat{T} = -\hat{p}$[38]. See Supplementary Note 7 for the details. Note that the detector state is described by the ancillary state with transformation by the nonlinear feedforward operations. For a reference of the detector states, we also test the case of using vacuum ancillary states instead of the non-Gaussian states because vacuum states are the most experimentally feasible ideal Gaussian states. Note also that Eq. (3) is consistent with Heisenberg picture Eq. (1) as the detector state $\hat{\Pi}_m$ would be proportional to the eigenstate of $\hat{p} + \gamma\hat{x}^2$ with eigenvalue $m$ with an ideal ancillary CPS.

The detector states can be reconstructed by detector tomography from conditional probabilities $\mathrm{prob}(m|\alpha) = \mathrm{Tr}[\hat{\Pi}_m |\alpha\rangle\langle\alpha|]$, using the set of coherent states $\{|\alpha\rangle\}$ forming an overcomplete basis for Hilbert space of the system, and the iterative maximum likelihood analysis[39] (see Supplementary Note 4). This is also advantageous experimentally as the coherent states are resilient to losses and have been already employed for tomography of homodyne and photon number resolving detectors[40,41].

Figure 3 shows the normalized Wigner functions of the detector states. Figure 3a and b show the ideal detector states, calculated according to Eq. (3), for the vacuum and the experimental non-Gaussian ancilla with $\mathrm{var}(\hat{p} + \gamma\hat{x}^2) = 0.64$ and $\mathrm{var}(\hat{p} + \gamma\hat{x}^2) = 0.56$, respectively. Figure 3c and d then show detector states for vacuum and actual non-Gaussian ancilla reconstructed from experimental data. The fidelities between the theoretically calculated detector states and experimentally reconstructed ones are 0.992 for the vacuum ancillary states and 0.993 for the non-Gaussian ancillary states. Detector states

for measured values $m$ differ simply by displacement in $p$, which is consistent with the analysis of the first moments. Although reconstructed detector states are noisy because of limited number of data points (see Supplementary Note 6), the averaged detector states show the significant properties of the nonlinear quadrature measurement in Fig. 3. Note that the averaging and renormalizing are done after $p$-displacement by measured results $m$ to cancel the intrinsic displacement of the detector states. The parabolic shape is qualitative evidence of the nonlinear quadrature measurement. It is determined by the nonlinear feedforward and therefore present for both kinds of ancillary states. The main difference is that the non-Gaussian ancilla leads to detector states with variance of the nonlinear quadrature operator $\hat{p} + \gamma\hat{x}^2$ equal to $0.67 \pm 0.01$ in Fig. 3d. In contrast, the vacuum state produces variance equal to $0.74 \pm 0.01$ (Fig. 3c). Note that the mean and standard error are calculated via a bootstrap method. This means the excess noise of the measurement with non-Gaussian ancilla, constructed as a superposition of the vacuum and single photon states, is already suppressed by 10% relative to the vacuum level, which is consistent with the measured nonlinear squeezing of the ancillary state. Thus we can see that even small nonlinear squeezing of the ancilla can already provide an observable effect.

## Discussion
We have implemented a nonlinear quadrature measurement of $\hat{p} + \gamma\hat{x}^2$ using the nonlinear electro-optical feedforward and non-Gaussian ancillary states. The nonlinear feedforward makes the tailored measurement classically nonlinear, while the ancillary state pushes the measurement into highly non-classical regime and determines the excess noise of the measurement. By using a non-Gaussian ancilla we

have observed 10% reduction of the added noise relative to the use of vacuum ancillary state, which is consistent with the amount of nonlinear squeezing in the ancilla. Higher reduction of the noise can be realized in the near future by a better approximation of the CPS using a superposition of higher photon number states[38,42]. We can now create broadband squeezed state of light beyond 1 THz[8,9] and can make a broadband amplitude measurement on it with 5G technology beyond 40 GHz[10], as well as a broadband photon-number measurement beyond 10 GHz[11]. Furthermore, the nonlinear feedforward presented here can be compatible with these technologies if an application specific integrated circuit (ASIC) is developed based on the FPGA board presented here. By using such technologies we can efficiently create non-Gaussian ancillary states with large nonlinear squeezing by heralding schemes[36,43] even when the success rate is very low. It is because we can repeat heralding beyond 10 GHz and can compensate for the very low success rate.

When supplied with such high-quality ancillary state, this nonlinear measurement can be directly used in the implementation of the deterministic non-Gaussian operations required in the universal quantum computation. Our experiment is a key milestone for this development as it versatilely encompasses all the necessary elements for universal manipulation of the cluster states. Furthermore, this method is extendable to multiple ancillary states case in implementation of the higher-order quantum non-Gaussianity[44] and multimode quantum non-Gaussianity[45].

Our experiment demonstrates an active, flexible, and fast nonlinear feedforward technique applicable to traveling quantum states localized in time. If the nonlinear feedforward system is combined with the cluster states[13,14] and GKP states[19], all operations required for large-scale fault-tolerant universal quantum computation can be implemented in the same manner. As such, we have demonstrated a key technology needed for optical quantum computing, bringing it closer to reality.

## Methods

### Details of optical setups

The light source of this experiment is a continuous-wave Ti:sapphire laser with a wavelength of 860 nm. The light is divided into three parts. First, one of the beams is used to pump a second harmonic generator (SHG), which is a bow-tie shaped cavity with 500 mm roundtrip with a periodically-poled lithium niobate (PPLN) crystal inside as a nonlinear medium for SHG. The generated beam from SHG with a wavelength of 430 nm pumps an optical parametric oscillator(OPO).

Second part is used for local oscillator (LO) beams of homodyne detectors, passing through two acousto-optic modulators (AOMs) and a mode cleaning cavity (MCC). The output beam from the MCC is distributed to two homodyne detectors. One of the local oscillator beams is coupled to a waveguide electro-optic modulator (EOM) for phase rotation by the feedforward operation. Displacement beam for the idler mode is also picked from this beam.

The last part is used for controlling the optical path, i.e., for cavity locking (locking beams) and for phase locking (probe beams). Frequencies and amplitudes of the control beams are controlled by AOMs. The frequencies of each beam are differently shifted for phase locking, where we actively feedback and control the phases of light to synchronize beat notes of interference to reference signals. The frequency shifts are 120 kHz for the probe of the idler mode, 5.5 MHz for the probe of the input beam. Locking beam of the OPO is also detuned by 1 MHz. The modulation signals and reference signals are generated by synchronized direct-digital synthesizers. The control beams are switched on and off periodically, which is called as sample and hold technique. In the sample phase, the control beams are turned on and we activate the feedback controls of phase and cavity locking. In the hold phase, the control beams are turned off and we deactivate the feedback, keeping the condition of the optical system. This technique

is used to avoid the control beams to disturb photon detection as large fake counts and to saturate the homodyne detectors.

The setup for generating the ancillary state is the same as ref. 35. The OPO used in this experiment is a triangle cavity with 108 mm round trip formed by two spherical mirrors and one plate polarizing beamsplitter (PBS). Inside the cavity, a type-II periodically-poled potassium titanyl phosphate (PPKTP) crystal with 20 mm long is put between two spherical mirrors. One of the spherical mirrors is an output coupler with a transmittance of 14%. This OPO is called an asymmetric OPO because the cavity is single resonant in the polarization to make the wave packet shape of the signal state into an exponentially rising shape for real-time quadrature measurement[46]. The pump beam is enhanced by a buildup cavity around the OPO. The asymmetric OPO is pumped and generates a two-mode squeezed state in two orthogonal polarization. The idler mode is s-polarized and resonant to the OPO while signal mode is p-polarized and does not resonant. The idler mode is displaced via interference of a coherent state at a beam splitter of 99% reflectivity. Frequency filters put on the idler path are designed as Fabry-Perot cavities with linewidths of 140.1 MHz and 90.9 MHz, respectively. The filtered idler beam is sent to an avalanche photodiode (APD; Excelitas technologies, SPCM-AQRH-16), and the click heralds the generation of non-Gaussian ancillary state.

The non-Gaussian ancillary state is generated in a wave packet localized in time domain and is compatible with time-domain multiplexing technique. Thus, we have to synchronize the feedforward system with the arrival of the wave packet accompanied by a heralding signal. To compensate the delay of the electrical trigger of the heralding signal compared to the arrival timing of optical wave packet (which is occurred by asymmetric optical path lengths, latency of the APD, and latency of cables from the APD to the FPGA board), 8.4 meters optical delay line is put on the signal path of the ancillary state. The delay line includes a Herriott cell, whose two spherical mirrors (whose curvature radius is $R = 1000$ mm) face each other at 168.5 mm distance. One of the mirrors has a hole through it to inject and output the light. The light is injected into the cell and go back and forth between the mirrors 16 times, and is output from the cell after about 28 ns.

The beam splitter used for interference of the input state with the ancillary state is a variable beam splitter that consists of two plate PBS and a half wave plate. The transmittance of this beam splitter is set to 50% during the experiment of nonlinear measurement, while it is set to about 100% during the characterization of the ancillary state. Another delay line synchronizes the optical signal and electrical signal for nonlinear feedforward operation. This delay line is implemented by a Herriott cell with two spherical mirrors ($R = 1000$ mm) at the distance of 645 mm. The length of optical path of the delay line is about 17.3 m corresponding to 64 ns.

### Design of feedforward circuits

The feedforward circuits play two roles. One is to extract quadrature information from measurement outcomes of a homodyne detector, and the other is to calculate a nonlinear function of the quadrature.

To extract the quadrature of a real-valued temporal mode from a homodyne measurement with a continuous-wave local oscillator, we have to integrate the measurement outcomes weighted with the mode function. This calculation can be processed in real-time with passive system if the impulse response of the measurement devices is designed as time-reversal of a desired mode function. Hence, this technique is called as real-time quadrature measurement and used in a few researches of quantum states[35,46,47]. For that purpose, we construct the circuits mainly with broadband and flat frequency response components, and add a low-pass filter which determines the shape of impulse response. While the quantum state is localized in the wave packet with the bandwidth of about 35 MHz, we use

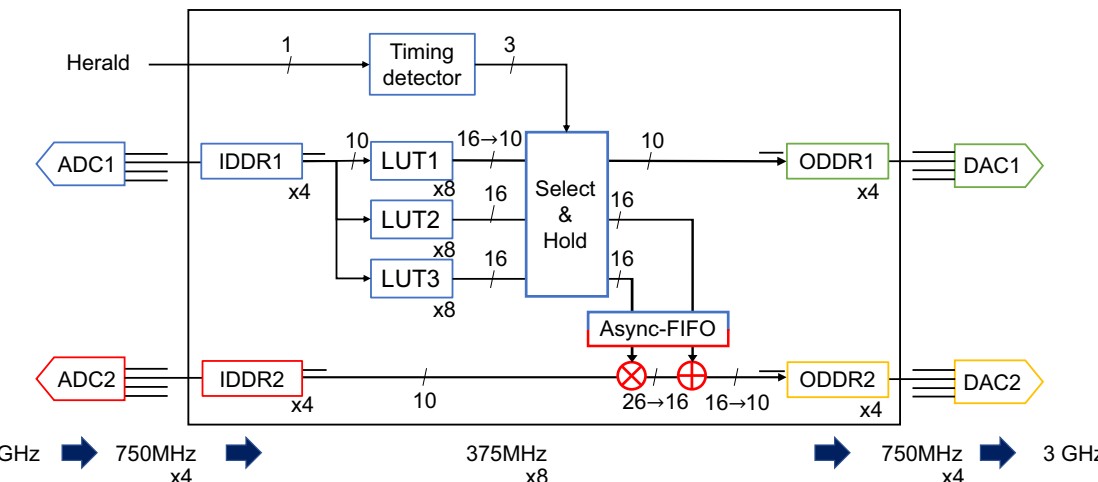

**Fig. 4 | Schematic diagram of the FPGA configuration.** The color of each component shows the clock domain. Numbers on the signal lines show the number of bit width. Three LUTs are implemented for future work. In this experiment, LUT1 is used for the nonlinear calculation. ADC analog-to-digital converter, IDDR input-double-data-rate primitive, LUT looking-up table, ODDR output-double-data-rate primitive, DAC digital-to-analog converter, Async-FIFO asynchronized fast-in-fast-out primitive.

homodyne detectors with about 200 MHz flat bandwidth, DC-coupled amplifiers and offset controller with about 1 GHz bandwidth. The low-pass filter is the same as the one used in ref. 35, and has the frequency response corresponded to the asymmetric OPO and the filtering cavities.

To calculate the nonlinear function of the quadrature, we employ a low-latency FPGA board (Fig. 4). The board is equipped with two analog-to-digital converters (ADCs), two digital-to-analog converters (DACs), and a field programmable gate array (FPGA) for signal processing. The ADCs (EV10AS152A, Teledyne e2v) are synchronized with a 3 GHz sampling clock. The output signal (10 bits resolution) is deserialized to 8 parallel channels inside the FPGA (Kintex-7 325T, Xilinx) with a 375 MHz processing clock via ADCs themselves and input-double-data-rate (IDDR) primitive of the FPGA. The DACs (EV10DS130AG, Teledyne e2v) also runs at 3 GHz, serializing the 8 parallel channels via output-DDR(ODDR) primitives of FPGA and DACs themselves. An essential property of the ADC and DAC is low latency, where the pipeline delay of the ADC and the DAC are 7.5 clock cycles and 4.5 clock cycles, respectively. Analog parts of the FPGA board has about 450 MHz − 1dB bandwidth, which is enough broad to treat the signal from a homodyne detector.

Looking-up tables for the nonlinear calculation in the FPGA board are implemented by block random access memories (BRAMs). Precomputed values of the nonlinear function are loaded to the BRAMs by a soft microprocessor core. The output signal of the FPGA board is normally turned off. A heralding signal of the ancilla preparation turns on the feedforward operation, holding the results of nonlinear calculation until the end of wave packet of the ancillary state. Since the quadrature signals are deserialized to 8 parallel channels inside the FPGA, we have to take care of the timing for triggering nonlinear feedforward. Because of the 8 parallel channels, if we employ conventional strategy, jitters up to 2.67 ns occur. In this work, however, we implement a time-to-digital converter for the trigger signal via a tapped delay lines, to cancel this jitter.

The output signal of the FPGA board is amplified to drive the EOM. The gain is tuned so that the range of the output voltage is the half-wave modulation voltage of the EOM to maximize the resolution of the phase rotation.

## Data availability
The data generated in this study are deposited in the RIKEN repository under accession code [https://dmsgrdm.riken.jp/3gvw2/].

## Code availability
All codes used in this work for data analysis and simulations are available from the corresponding authors upon request.

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

## Acknowledgements

This work was partly supported by Japan Science and Technology Agency (Moonshot R&D) Grant No. JPMJMS2064, Japan Society for the Promotion of Science KAKENHI Grant No. 18H05207 and No. 21J11615, the UTokyo Foundation, and donations from Nichia Corporation. F.H. acknowledges supports from the Forefront Physics and Mathematics Program to Drive Transformation (FoPM). W.A. acknowledges supports from the Research Foundation for Opto-Science and Technology. H.Y. acknowledges the Australian Research Council Centre of Excellence for Quantum Computation and Communication Technology (Project No. CE170100012). P.M. acknowledges Grant No. 22-08772S of Czech Science Foundation (GACR) and the European Union's HORIZON Research and Innovation Actions under Grant Agreement no. 101080173 (CLUSTEC). R.F. acknowledges the project 21-13265X of Czech Science Foundation. P.M. and R.F. acknowledges EU H2020-WIDESPREAD-2020-5 project NONGAUSS (951737) under the CSA - Coordination and Support Action.

## Author contributions

A.S., S.K., and H.O. designed and built the experimental setups supported by K.T., and J.Y. for debugging. E.H. designed the FPGA board used for the feedforward. A.S. and F.H. acquired and analyzed the data with discussions with W.A., P.M., R.F. and other co-authors. P.M. and R.F. led the theoretical simulations. A.F supervised the project. A.S. wrote the manuscripts with assistance from W.A., P.M., R.F., H.Y. and other co-authors.

## Competing interests

The authors declare no competing interests.
