## [Peer Review File · Nature Communications]

Nonlinear feedforward enabling quantum computationREVIEWER COMMENTS

Reviewer #1 (Remarks to the Author):

I have read the paper titled «Nonlinear feedforward enabling quantum computation». The paper proposes a method for non-Gaussian measurements of traveling light. I believe that the paper is of great importance for the development of continuous variable quantum information processing.

I think this result deserves publication in Nature Communications after minor edits have been made to the paper.

Below are my two minor comments, which are not related to the quality of the work, but only to the text of the article.

1) The authors of the work demonstrated a way to carry out non-Gaussian measurements by projecting the input state onto the states of the cubic phase. In the paper, the authors say that these measurements are necessary for universal continuous-variable quantum computation. However, the authors do not mention other useful protocols where such measurements are needed. For example, continuous-variable teleportation protocols, where, through the use of non-Gaussian transformations, fidelity can be increased [Phys. Rev. A 104, 032420 (2021)]. Or protocols for generating states of Schrödinger's cats based on a cubic phase state [Physics Letters A 384 (2020) 126762]. In my opinion, in the introduction it would not be superfluous to mention such a wide applicability of this type of non-Gaussian states. This once again indicates the great significance of this work.

2) In Fig. 3, the authors show the states of the detector of non-Gaussian states they got. From the main text of the article, one gets the impression that only non-Gaussian states are studied in the paper as ancilla ones. Understanding why there are vacuum states on the graphs comes only after reading the auxiliary materials. I believe the authors should describe in the main text a little more the case when the vacuum state is used, and when it is non-Gaussian.

I believe that the presented work will be useful for a broad audience of quantum information theorists and experimentalists.

Reviewer #2 (Remarks to the Author):

The paper demonstrated a fast and flexible nonlinear feedforward that allows the essential measurement required for fault-tolerant and universal quantum computation. Up to my knowledge, that is the first time that such nonlinear feedforward is realized. There is no doubt that their results are important to the development of the field.

The experimental part is robust. There is a care in talking about the limitations and the results are consistent. I missed the equations that represent the states produced in the experiment and the processing of states starting from the input and auxiliary states until reaching the final state. I also did not see the estimated time spent on electronic processing in the FPGA and the delay time on the optical delay. I also did not see if the component $R(\theta(q))$ is a phase or amplitude modulator.

I would gladly suggest the paper acceptance, if the suggestions above are addressed.

Reviewer #1

I have read the paper titled «Nonlinear feedforward enabling quantum computation». The paper proposes a method for non-Gaussian measurements of traveling light. I believe that the paper is of great importance for the development of continuous variable quantum information processing.

I think this result deserves publication in Nature Communications after minor edits have been made to the paper.

We thank the reviewer for finding that the methods and experimental results in our manuscript are important for the development of continuous variable quantum information processing and deserving of publication.

1) The authors of the work demonstrated a way to carry out non-Gaussian measurements by projecting the input state onto the states of the cubic phase. In the paper, the authors say that these measurements are necessary for universal continuous-variable quantum computation. However, the authors do not mention other useful protocols where such measurements are needed. For example, continuous-variable teleportation protocols, where, through the use of non-Gaussian transformations, fidelity can be increased [Phys. Rev. A 104, 032420 (2021)]. Or protocols for generating states of Schrödinger's cats based on a cubic phase state [Physics Letters A 384 (2020) 126762]. In my opinion, in the introduction it would not be superfluous to mention such a wide applicability of this type of non-Gaussian states. This once again indicates the great significance of this work.

We thank the reviewer for pointing this out. We agree that the non-Gaussian measurement and a cubic phase gate implemented by this measurement have been considered as essential resources for various continuous variable protocols including probabilistic process. Regarding this comment, we have added a sentence in the introduction part.

2) In Fig. 3, the authors show the states of the detector of non-Gaussian states they got. From the main text of the article, one gets the impression that only non-Gaussian states are studied in the paper as ancilla ones. Understanding why there

are vacuum states on the graphs comes only after reading the auxiliary materials. I believe the authors should describe in the main text a little more the case when the vacuum state is used, and when it is non-Gaussian.

We thank the reviewer for pointing this out. We have added explanation in the third paragraph of the “Tomography of tailored measurement” section.

Reviewer #2

The paper demonstrated a fast and flexible nonlinear feedforward that allows the essential measurement required for fault-tolerant and universal quantum computation. Up to my knowledge, that is the first time that such nonlinear feedforward is realized. There is no doubt that their results are important to the development of the field.

We thank the reviewer for summarizing our manuscript clearly and finding the novelty and significance of our manuscript.

I missed the equations that represent the states produced in the experiment and the processing of states starting from the input and auxiliary states until reaching the final state.

We thank the reviewer for pointing this out. Regarding the comment, we have added explanation in the Schrodinger picture in the main text. Please refer to the third paragraph in the “Tomography of tailored measurement” section.

I also did not see the estimated time spent on electronic processing in the FPGA and the delay time on the optical delay.

Although we estimate the time spent inside the FPGA is about 10-15 ns (depending on the definition of “inside/outside”), most of the time is spent just for signal propagation in the FPGA and the genuine latency required for the nonlinear calculation is one clock cycle, 2.67 ns. The total latency from the analog input to the analog output of the FPGA board is about 26.8 ns, including inevitable additional delays caused by analog circuits, pipelining in the AD/DA converters, etc.

I also did not see if the component $R(\theta(q))$ is a phase or amplitude modulator.

We apologize for the confusion in the original manuscript. We have clarified the definition of $R(\theta(q))$ as follows, “via phase rotation $R(\theta(q))$ of the local oscillator beam of HD2.”

Editorial comments

After the referees submitted their reports, we have had additional exchanges with referee #2, mainly concerning the degree of overlap with the recent work "Nonlinear Squeezing for Measurement-Based Non-Gaussian Operations in Time Domain" (Phys. Rev. Applied 15, 024024), ref.[33] in the manuscript. Both we and the referee are concerned by the degree of advance constituted by the current manuscript in the light of this previous achievement, and would like to consider your response to this concern (as well as the other ones raised in the reports) before we make a final decision on publication.

We would like to emphasize that there is a big difference between the achievements in ref. [33] and that in the current manuscript. A crucial difference comes between nonlinear squeezing generation and its direct use as a resource in a quantum measurement with nonlinear feedforward control. The difference between them is similar to generation of entanglement and the realization of quantum teleportation, for example. Although the entangled states are an essential element, entire linear feedforward operations depending on the outcomes of Bell measurement are necessary to demonstrate deterministic quantum teleportation. What we have achieved in the current manuscript is like that feedforward, while only the preparation of an ancillary state for a cubic phase gate was demonstrated in ref. [33]. To be sure that point is not missed, we emphasize it more in the revised manuscript.

The necessity of feedforward would be clearer if we consider the case without nonlinear feedforward. Even though the ancillary state is crucial to obtaining quantum non-Gaussianity, we could not measure the nonlinear quadrature without nonlinear feedforward. What we could do without nonlinear feedforward is just conditioning of the nonlinear quadrature, by post-selecting measurement outcomes only to be $p + \gamma x^2 = 0$. This only probabilistic gate cannot be used in measurement-induced quantum computation. Moreover, for any other applications of such probabilistic measurements, the success rate of this conditioning, which depends on the input and ancillary states, becomes lower if the ancillary states get closer to the ideal cubic phase state. With the nonlinear feedforward, we do not suffer from this limitation of the success rates and reach the measurement strategy necessary for measurement-induced quantum computation and other possible applications.

In addition, various types of nonlinear feedforward can be implemented with our setup beyond the cubic phase gate since our system has the ability of flexible signal processing with a constant latency.

Additionally, we would be grateful if you could include more information about how

the operation transforms the input state, and in particular about the amount of non-gaussianity imparted to the input state or the amount of Wigner-function negativity created. Also, including the fidelity of the experimental process compared to a useful target process would be useful.

As for the information about the experimental process, we have added explanations in Schrodinger picture, as mentioned in the response to Reviewer #2.

The value of the Wigner negative volume (integration of the negative value of the Wigner function over the whole phase space) for the averaged detector state is about 0.0063, including the reconstruction artifacts. The Wigner negative volume is limited by that of the non-Gaussian ancillary state, which is 0.0088. These values are minimal because the ancillary state is optimized for noise suppression in the nonlinear quadrature measurement, not for Wigner-function negativity, as discussed in ref. [33]. This situation is similar to approximated GKP states, which are also not optimized for the highest negative values of Wigner function but rather for periodical structures in the Wigner function. We believe the noise of measured value $p + \gamma x^2$ is a more suitable figure of merit for this measurement because more significant Wigner negativity does not necessarily mean lower noise of the nonlinear quadrature measurement. We need a cubic-type quantum non-Gaussianity for this measurement, which seems different from the photon number parity.

Regarding the accuracy of the experimental process, we calculate the fidelity between the theoretical and actual estimated detector state shown in Fig.3 and added the value to the discussion. Although the values exceed 0.99, we stress that fidelity is only an average measure suitable if we treat pure states and operations, and that fidelity for mixed quantum states tends to be higher than that for pure quantum states. In addition, since the detector state of the ideal $\hat{p} + \gamma \hat{x}^2$ measurement is unphysical as same as ideal cubic phase states, it does not bring relevant information about the specific features of our non-Gaussian measurement even in the ideal situation. This is also similar as it is, for non-Gaussian GKP-like states and ideal GKP states.

REVIEWERS' COMMENTS

Reviewer #2 (Remarks to the Author):

I'm satisfied with all clarifications and corrections done.

I believe the manuscript should be published.

Reviewer #2(Remarks to the Author):

I'm satisfied with all clarifications and corrections done.

I believe the manuscript should be published.

We thank the reviewer again for your constructive comments.